# Trophic eggs affect caste determination in the ant *Pogonomyrmex rugosus*

Eléonore Genzoni[1], Tanja Schwander[1], Laurent Keller[1,2]*

[1]Department of Ecology and Evolution, Biophore Building, University of Lausanne, Lausanne, Switzerland; [2]Social Evolution Unit, Chesières, Switzerland

## eLife Assessment

This **important** manuscript by Genzoni et al. reports the striking discovery of a regulatory role for trophic eggs in ant caste determination. Prior to this study, trophic eggs were widely assumed to play only a nutritional role in the colony, but this **compelling** study shows that trophic eggs can suppress queen development, and therefore regulate caste determination in specific social contexts.

**Abstract** Understanding how a single genome creates distinct phenotypes remains a fundamental challenge for biologists. Social insects provide a striking example of polyphenism, with queen and worker castes exhibiting morphological, behavioral, and reproductive differences. Here, we show that trophic eggs, which do not contain an embryo and are primarily regarded as a source of food, play a role in the process of caste determination in the harvester ant *Pogonomyrmex rugosus*. When first instar larvae were given access to trophic eggs, they mostly developed into workers. By contrast, larvae without access to trophic eggs developed into queens. We found that trophic eggs differ in many ways from viable eggs, including texture, morphology, and their contents of protein, triglycerides, glycogen, sugar, and small RNAs. Moreover, comparison of miRNA fragment size distributions suggests differences in the composition of miRNAs between the two egg types. This is the first demonstration of trophic eggs playing a role in caste determination in social insects.

*For correspondence:
laurent.keller01@gmail.com

## Introduction

Many species of insects, spiders, amphibians, marine invertebrates, and sharks produce trophic eggs, a special type of eggs that do not contain an embryo (*Levin and Bridges, 1995*; *Blake and Arnofsky, 1999*; *Collin, 2004*; *Kudo and Nakahira, 2004*; *Perry and Roitberg, 2006*; *Strathmann and Strathmann, 2006*; *Gibson et al., 2012*; *López-Ortega and Williams, 2018*). It is generally assumed that these non-developing eggs are either a by-product of failed reproduction or that they serve as nutrition for offspring (*Perry and Roitberg, 2006*). However, the suggestion that trophic eggs solely provide a nutritional function is based on surprisingly little evidence. We here report a direct function of trophic eggs in the determination of alternative phenotypes in ants.

Trophic eggs have been reported in many ant species (*Figure 1* and *Supplementary file 1, table 1*) and are generally thought to mostly or only serve as food for offspring (*Crespi, 1977*; *Hölldobler and Wilson, 1990*). Trophic eggs play an important role during the time of colony founding (*Hölldobler and Wilson, 1990*). Because in most ant species, queens do not forage after the mating flight, they metabolize the alary muscles and fat bodies and convert them into eggs, which serve as food to rear the first batch of larvae (*Huber, 1905*; *Keller and Passera, 1989*). Except for a few species such as *Crematogaster smithi* (*Heinze et al., 1995*) and *Acanthomyrmex ferox* (*Gobin and Ito, 2000*), the queens generally stop producing trophic eggs after the eclosion of the first workers (see *Figure 1* and

**Figure 1.** Trophic egg production is widespread in ants. Simplified phylogenetic tree of ant subfamilies redrawn after *Romiguier et al., 2022*. The number of species with documented trophic egg production by queens, workers, or both castes is indicated for each subfamily. The question mark indicates that it is unclear whether trophic eggs can be produced by queens (in *Lasius niger*, trophic eggs are produced by workers and possibly queens, see *Supplementary file 1, table 1*). Details on the species and related references can be found in *Supplementary file 1, table 1*.

*Supplementary file 1, table 1*). So far, the absence of trophic eggs has been reported in only one species (*Supplementary file 1, table 1*).

In many ant species, workers are capable of producing haploid males but lack a spermatheca and the ability to produce diploid female offspring (*Hölldobler and Wilson, 1990*; *Bourke, 1988*; *Hammond and Keller, 2004*; *Wenseleers and Ratnieks, 2006*). In some species, workers further lay trophic eggs (see *Supplementary file 1, table 1*). These eggs have a trophic function, in particular in genera such as *Pogonomyrmex* where there is no or only little trophallaxis (regurgitative food sharing) among workers. In such species, it has been suggested that trophic eggs may play an important role in food distribution within the colony (*Gobin and Ito, 2000*).

Several studies suggested that the presence of trophic eggs may affect the developmental trajectory of larvae. A study in the Argentine ant *Linepithema humile* showed that the presence of queens in colonies was associated with a drastic decrease in the number of worker-laid trophic eggs as well as a decrease in the proportion of larvae developing into queens (*Bartels, 1988*). Bartels thus proposed that trophic eggs may increase the probability of larvae to develop into queens, which are much larger than workers (*Bartels, 1988*). In some lineages of *Pogonomyrmex barbatus*, queens laid a higher proportion of trophic eggs upon the experimental increase of maternal juvenile hormone (*Helms Cahan et al., 2011*). Because the treatment also strongly reduced the number of workers produced but triggered a 50% increase in worker body size, trophic eggs were suggested to affect worker size. An increase in the proportion of trophic eggs has also been suggested to be associated with an increase in the proportion of larvae developing into queens (*L. humile*: *Bartels, 1988*; *P. barbatus*: *Helms Cahan et al., 2011*). Finally, because they observed an increase in nitrogen content with increasing female caste size in the ant *Pogonomyrmex badius*, *Smith and Suarez, 2010* suggested that the larger castes may consume more trophic (nutritional) eggs than the smaller caste, which would feed more on foraged insects. In summary, in at least three species of ants, trophic eggs may play a role in the developmental trajectories of female larvae.

While conducting egg cross-fostering experiments in the ant *Pogonomyrmex rugosus* to study worker size variation, we observed a sudden increase in the frequency of females developing into queens. During these experiments, we discarded trophic eggs and only cross-fostered eggs that would eventually hatch (hereafter viable eggs). This raises the possibility that the absence of trophic eggs influenced the process of caste determination. These observations prompted us to investigate

**Table 1.** Wald–Wolfowitz runs tests on the queen's egg sequence.
Significant p-values (corrected for multiple testing) indicate that queens do not lay viable and trophic eggs in a random sequence.

| Queen ID | p-value for egg sequence | p-value of random sequence | Number of eggs per sequence |
|---|---|---|---|
| 338 | $4.1 \times 10^{-11}$ | 0.419 | 94 |
| 117 | $4.3 \times 10^{-07}$ | 0.567 | 63 |
| 173 | $1.7 \times 10^{-03}$ | 0.755 | 92 |
| 303 | $4.8 \times 10^{-13}$ | 0.765 | 110 |
| 215 | $9.8 \times 10^{-11}$ | 0.292 | 70 |
| 120 | $1.4 \times 10^{-05}$ | 0.518 | 75 |
| 12B | $1.9 \times 10^{-03}$ | 0.298 | 38 |
| 316 | $3.4 \times 10^{-09}$ | 0.737 | 93 |
| 193 | $4.3 \times 10^{-12}$ | 0.655 | 101 |
| 150 | $1.4 \times 10^{-10}$ | 0.630 | 62 |
| 125 | $1.5 \times 10^{-05}$ | 0.404 | 58 |

whether trophic eggs play a role in caste determination in *P. rugosus*. Our experiments revealed that the presence of trophic eggs reduces the probability that female larvae develop into reproductive individuals. Metabolomic analyses also revealed profound differences between viable and trophic eggs, including in the composition of miRNAs and content of protein, triglycerides, glycogen, and sugar.

## Results

### Trophic and viable egg characteristics

*P. rugosus* queens lay two types of eggs that are morphologically different. Viable eggs are white with a bright surface and have a distinct oval shape, a homogenous content as well as a solid chorion (*Figure 2A*), while trophic eggs are rounder, have a smooth surface and a granular-looking content as well as a fragile chorion (*Figure 2D*). Trophic eggs had a significantly larger volume (94.3 ± 4.3 nL; $n$ = 11) than viable eggs ($n$ = 14; 63.3 ± 1.6 nl; two-sample $t$-test, $t(23)$ = –9.54, p = $1.8 \times 10^{-09}$). While viable eggs showed embryonic development at 25 and 65 hr (*Figure 2B and C*) there was no such development for trophic eggs (*Figure 2E, F*). *P. rugosus* workers only laid viable eggs. They started to lay eggs approximately 3 weeks after queen removal ($n$ = 12 queenless recipient colonies), and approximately 90% of the eggs successfully hatched. However, only approximately 5% successfully developed into pupae which were all males.

The percentage of eggs that were trophic was higher before hibernation (61.6 ± 1.4% mean ± SE; $n$ = 43 colonies) than after (50.3 ± 2.0%; LMER, $t(86)$ = 5.04, p = $9 \times 10^{-6}$). The production of the two types of eggs was not random (Wald–Wolfowitz runs tests, p-values for the 11 queens in *Table 1*). Instead, each of the 11 queens tended to lay relatively long sequences of either viable (6.1 ± 0.7; mean number per sequence ± SE) or trophic eggs (6.0 ± 0.5; *Figure 3*).

The concentrations of protein, triglycerides, glycogen, and glucose were significantly higher in viable than trophic eggs (LMER, protein: $t$ = –13.11, p < 0.0001; triglycerides: $t$ = –11.66, p < 0.0001; glycogen: $t$ = –11.98, p < 0.0001; glucose: $t$ = –18.60, p < 0.0001; *Figure 4*).

The amount of small RNA (<200 nt, including miRNA and tRNA; *Nagano and Fraser, 2011*) was significantly higher in viable eggs (44.3 ± 1.4 ng, mean ± SE) than in trophic eggs (22.3 ± 1.1 ng; paired $t$-test, $t_{(23)}$ = 15.9, p = $6.5 \times 10^{-14}$). The same was true for longer RNAs (>200 nt; viable eggs: 7.6 ± 0.6 ng, mean ± SE; trophic eggs: 3.6 ± 0.3 ng; paired $t$-test, $t_{(23)}$ = 7.2, p = $2.7 \times 10^{-7}$).

The DNA quantification showed that the amount of DNA was about twice as high in viable (15.9 ± 1.9 ng/µl) than trophic eggs (8.8 ± 1.9 ng/µl; $t$-test, $t_{(4.7)}$ = 2.7, p = 0.045).

There was a significant difference in the miRNA fragment size distribution between viable and trophic eggs (Mantel test, $r_M$ = 0.26, p < 0.0001), as shown on the PCA (*Figure 5A*). There was no difference in the tRNA fragment size distribution between the two types of eggs (Mantel test, $r_M$ = 0.01, p = 0.30, *Figure 5B*).

## Trophic eggs influence caste fate of larvae

The percentage of larvae that developed into queens was significantly lower in recipient colonies that received trophic eggs (27 ± 9% mean ± SE; *n* = 22) than in recipient colonies without trophic eggs (83 ± 10%; *n* = 22; binomial GLMM (link='logit'), z = 4.25, p = 2 × 10⁻⁵; *Figure 6A* and *Supplementary file 1, table 2*). The survival of larvae until the pupal stage was also significantly lower in the colonies without trophic eggs (16.9 ± 3.8%; *n* = 22; LMER, z = 2.66, p = 0.008) than in colonies with trophic eggs (30.2 ± 6.7%; mean ± SE; *n* = 22), but the 1.8-fold survival decrease cannot fully account for the threefold difference in queen percentage between the two treatments. Furthermore, there was no significant correlation between larval mortality and the percentage of larvae developing into queens (*n* = 44 recipient colonies; LMER, z = 0.97, p = 0.34; *Figure 6B*). These analyses allow us to exclude differential survival between castes as the sole explanation for the higher percentage of queens developing in the recipient colonies without trophic eggs.

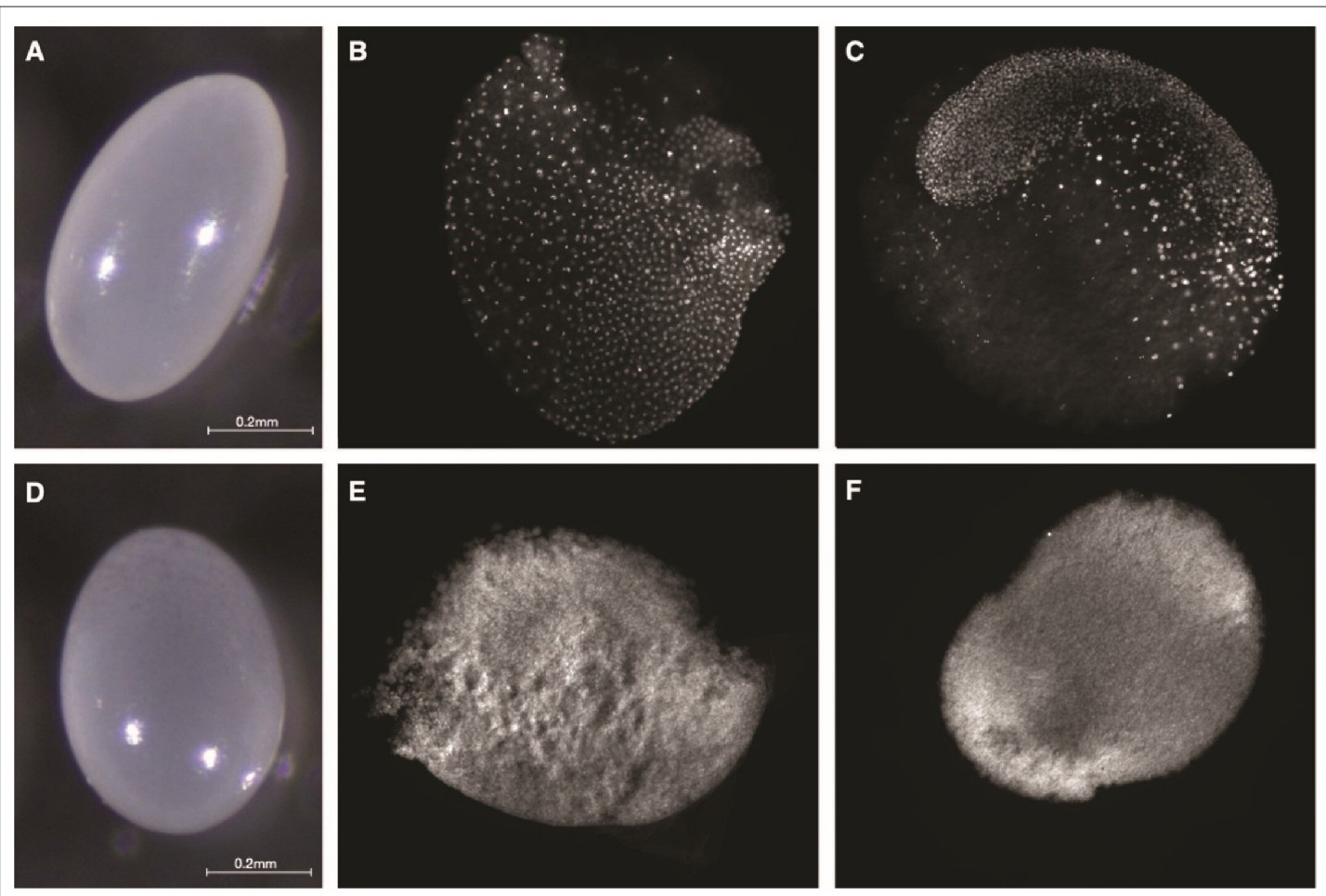

**Figure 2.** Morphology and embryonic development of eggs laid by *P. rugosus* queens. Morphology of viable (**A**) and trophic (**D**) eggs. Fluorescence images with DAPI-counterstained nuclei showing embryonic development of viable eggs at approximately 25 hr (**B**) and 65 hr (**C**). For trophic eggs, there was no embryonic development at 25 hr (**E**) nor at 65 hr (**F**).

## Discussion

Our study reveals that *P. rugosus* queens lay a very high proportion (0.6) of trophic eggs. These eggs differ in many ways from viable eggs. First, trophic eggs are larger, rounder, have a smoother surface, a more granular-looking content as well as a more fragile chorion than viable eggs. Similar differences between trophic and viable eggs have been reported in other ant species (*Wilson, 1976*; *Wardlaw and Elmes, 1995*; *Gobin et al., 1998*; *Dietemann and Peeters, 2000*; *Dietemann et al., 2002*; *Perry and Roitberg, 2006*; *Lee et al., 2017*). Our analyses also showed that trophic eggs are solely laid by queens; *P. rugosus* workers are able to produce viable eggs which occasionally develop into males, but they do not lay trophic eggs. Moreover, trophic eggs have a reduced DNA content.

Importantly, our experiments showed that the presence of trophic eggs influences the process of caste determination. First instar female larvae fed with trophic eggs were significantly more likely to develop into workers than larvae without access to trophic eggs. This was somewhat surprising because trophic eggs are generally thought to be an important source of nutrients to the colony and, everything else being equal, one would think that eating such eggs should increase the likelihood of females to develop into queens (which are usually larger than workers). Indeed, two earlier studies suggested that an increase in the proportion of trophic eggs might be associated with an increase in the proportion of larvae developing into queens (*L. humile*: *Bartels, 1988*; *P. barbatus*: *Helms Cahan et al., 2011*). However, in these two studies, variation in the availability of trophic eggs was associated with other differences (number of queens in the colony, *Bartels, 1988*; Administration of a JH analog, *Helms Cahan et al., 2011*), making it difficult to determine what factor had a causal effect. It would be interesting to experimentally manipulate the quantity of trophic eggs in *L. humile* and *P. barbatus* to determine whether they have a positive or inhibitory effect on the likelihood of larvae to develop into queens.

Our analyses revealed that trophic eggs have a lower content of protein, triglycerides, glycogen, and glucose than viable eggs. A reduced protein content of trophic as compared to viable eggs has also been documented in *Pheidole pallidula* (*Lorber and Passera, 1981*). These findings are in line with the view that trophic eggs do not simply have a nutritional function, as it might then be expected that they should at least contain as much nutrients as viable eggs. Interestingly, our analyses also revealed important differences in RNA and miRNA content between the two egg types. miRNAs have already been suggested to influence larval caste determination in the honeybee (*Guo et al., 2013*) with worker jelly being enriched in miRNAs compared to royal jelly (*Guo et al., 2013*; *Zhu et al., 2017*). These studies suggest that it is not the royal jelly that stimulates larval determination into queen, but rather the worker jelly which stimulates the development of larvae into workers. Similarly, our study reveals that compounds found in trophic eggs, perhaps miRNAs, influence larval development toward the worker phenotype. Interestingly, it has also been recently shown that trophallactic fluid in the ant *Camponotus floridanus* contains non-digestive related proteins, microRNAs, and juvenile hormone (*LeBoeuf et al., 2016*). Moreover, comparison of trophallactic fluid proteins across social insect species revealed that many are regulators of growth, development, and behavioral maturation (*Meurville and LeBoeuf, 2021*). Finally, a recent study showed that pupae of several ant species produce secretions that play an important role for early larval nutrition, with young larvae

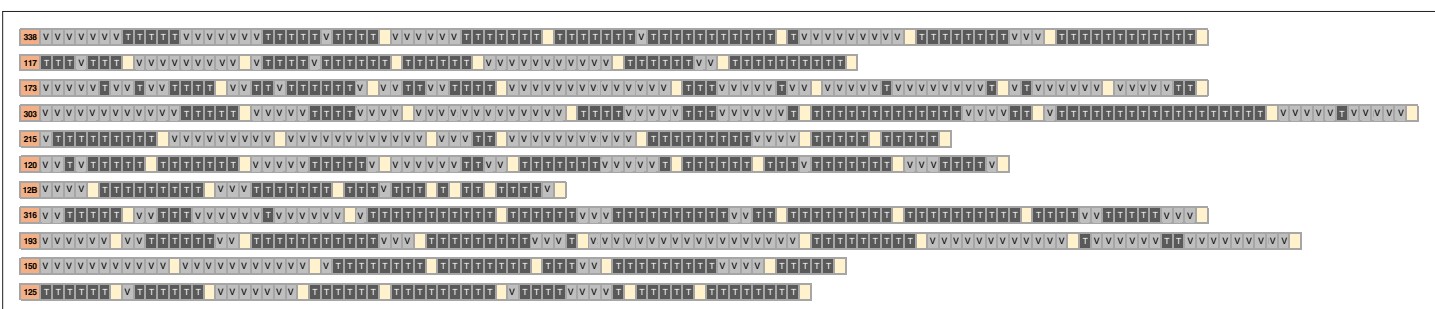

**Figure 3.** Egg-laying sequences from 11 *P. rugosus* queens. Every row shows the sequence of viable (V) and trophic (T) eggs laid by a given queen (queen ID in the orange cell). Each egg-laying session lasted 10 hr. The yellow squares indicate the intervals (16 hr to several days) between egg-laying sessions.

exhibiting stunted growth and decreased survival without access to the fluid (*Snir, 2022*). This raises the possibility that chemicals delivered in trophic eggs, trophallactic fluids, and pupae secretions play previously unsuspected roles in communication and caste development. Given that some ants do not perform trophallaxis, it would be interesting to determine whether there are differences in the content of trophic eggs of species performing trophallaxis and species that do not.

Maternal effects on the process of caste determination have been demonstrated in several social insect species, including *P. rugosus*, either by queen behavior or content of the eggs being produced (*de Menten et al., 2005*; *Linksvayer, 2006*; *Schwander et al., 2008*; *Libbrecht et al., 2013*; *Wei et al., 2019*). This is, to our knowledge, the first experimental demonstration that provisioning of trophic eggs influences caste fate. Since only queens produce trophic eggs in *P. rugosus*, trophic egg provisioning could be the main mechanism underlying the previously documented maternal effects on the process of caste determination. In species where workers produce trophic eggs (*Supplementary file 1, table 1*), the same mechanism could allow workers to influence colony level caste ratios if their eggs do also affect the process of caste determination. The presence of trophic eggs has been documented in only relatively few ant species (*Table 1*). However, their presence has been investigated in only a few species, and it is likely that trophic eggs are produced in most ant species, particularly in those with an independent mode of colony founding.

Finally, our analyses also revealed seasonal differences in the proportion of viable and trophic eggs, with a higher ratio of trophic eggs before hibernation than after. In *Pogonomyrmex*, the production of new queens occurs after hibernation (*Smith and Tschinkel, 2006*) or when the queen dies or is removed from the colony (pers. obs). Thus, new queens are typically produced when there are fewer trophic eggs. Our results predict that under natural conditions, a decrease in the proportion of trophic eggs should lead to an increase in the proportion of larvae developing into queens. The same logic applies to species where trophic eggs are laid only by the workers in queen-right colonies (*Supplementary file 1, table 1*). After the queen's death, workers start producing their own male offspring and lay mostly (if not only) viable eggs (*Temnothorax recedens*, *Dejean and Passera, 1974*; *Plagiolepis pygmaea*, *Passera, 1980*; *Myrmecia gulosa*, *Dietemann et al., 2002*), which again leads to a decrease, or cessation, in trophic egg production. A decrease in trophic egg production and the development of queens were observed simultaneously in freshly orphaned colonies of *T. recedens* (*Dejean and Passera, 1974*), *P. pygmaea* (*Passera, 1980*), and *M. gulosa* (*Dietemann et al., 2002*). These examples are consistent with the view that trophic eggs may also play a role in the process of caste determination in other ant species.

In addition to the seasonal differences reported in this study, previous studies in ants have also revealed that young queens produce more trophic eggs than older queens (*Hölldobler and Wilson, 1990*). Interestingly, the production of new queens in ant colonies starts only when colonies are several years old and have reached a relatively large size. Thus, this raises the possibility that the decrease of the proportion of trophic eggs laid by queens when they age may contribute to the production of new queens being restricted to colonies having reached a relatively large size.

Egg cannibalism has been reported in many ant species (*Wilson, 1971*; *Sorensen et al., 1983*; *Bourke, 1991*; *Crespi, 1977*; *Peeters and Tsuji, 1993*; *Aron et al., 1994*; *Heinze et al., 1999*; *Schultner et al., 2013*) and may serve many purposes, including preferential elimination of males, source of energy (*Sorensen et al., 1983*) and intracolony conflict (*Schultner et al., 2014*). Our study indicates a new potential role of egg cannibalism as a mechanism to regulate the allocation of energy into the production of new queens versus workers.

In conclusion, this study provides a new striking example of how females can influence the developmental fate of their offspring. Because many ants produce trophic eggs, it is possible that this mechanism of parental manipulation is widespread and plays an important role in the general process of caste determination. It would be interesting to conduct manipulative experiments similar to those of this study in additional species to determine whether trophic eggs broadly play a role in the process of caste determination. Of interest would also be to determine what chemicals in the egg are responsible for influencing the development of larvae.

## Materials and methods

Some populations of *P. rugosus* are characterized by a genetic caste determination system whereby development into queens or workers is determined by whether the eggs are fertilized by males of the

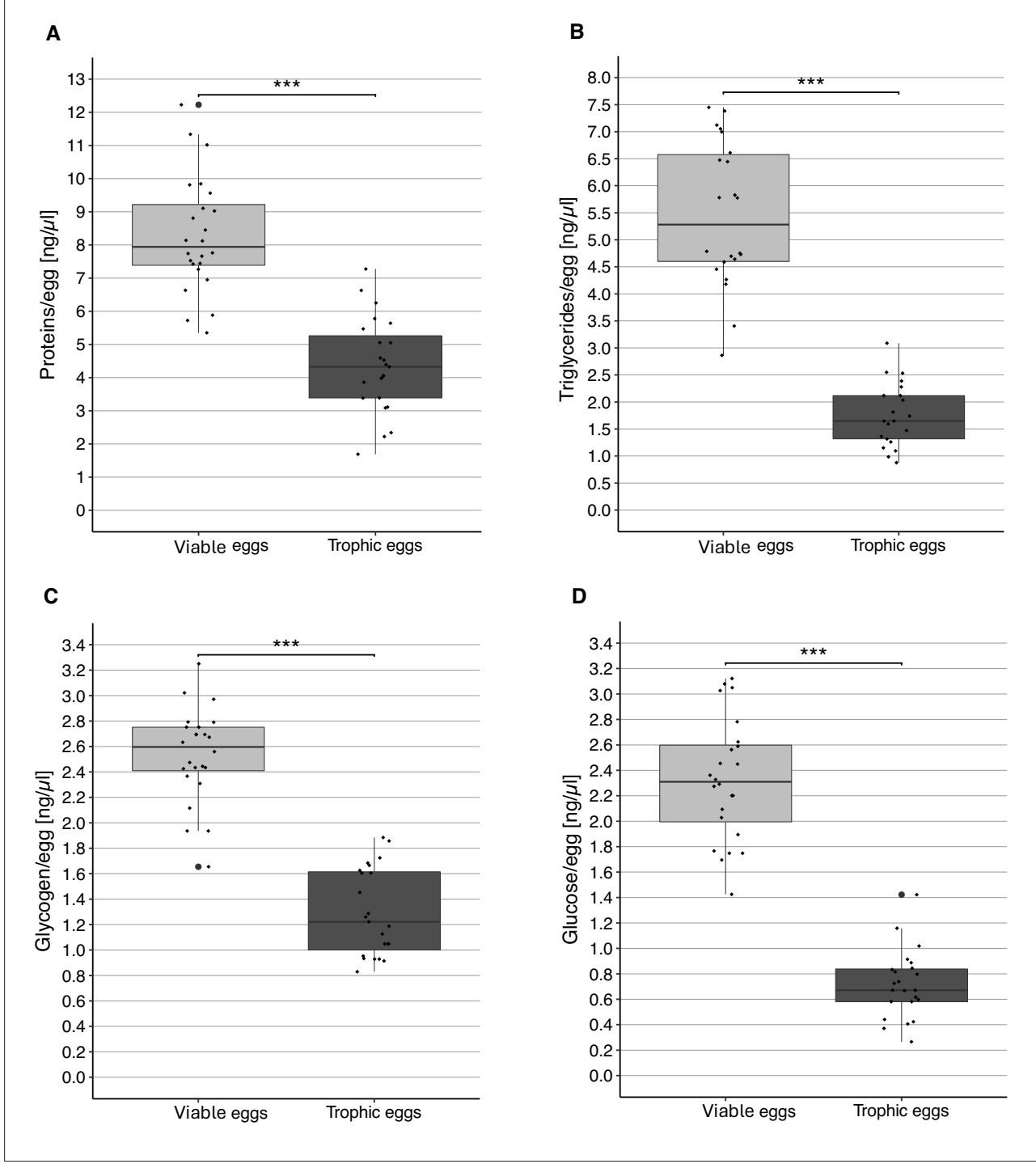

**Figure 4.** Content of viable and trophic eggs. Concentration (± standard error) of protein (**A**), triglycerides (**B**), glycogen (**C**), and glucose (**D**) in viable and trophic eggs. Each dot represents the average of the two replicates per colony. *** Indicates a p < 0.001.

same genetic lineage or a different genetic lineage than the queen producing the eggs (*Cahan et al., 2002*; *Julian et al., 2002*; *Volny and Gordon, 2002*; *Helms Cahan and Keller, 2003*). We collected queens from two populations (Bowie and Florence, Arizona, USA) known to harbor only colonies with non-genetic caste determination (*Schwander et al., 2007*). The queens were collected after mating flights in 2008 (Bowie) and 2023 (Bowie and Florence) to initiate colonies in the laboratory. The colonies were maintained in plastic boxes containing water tubes (glass tubes filled with water and

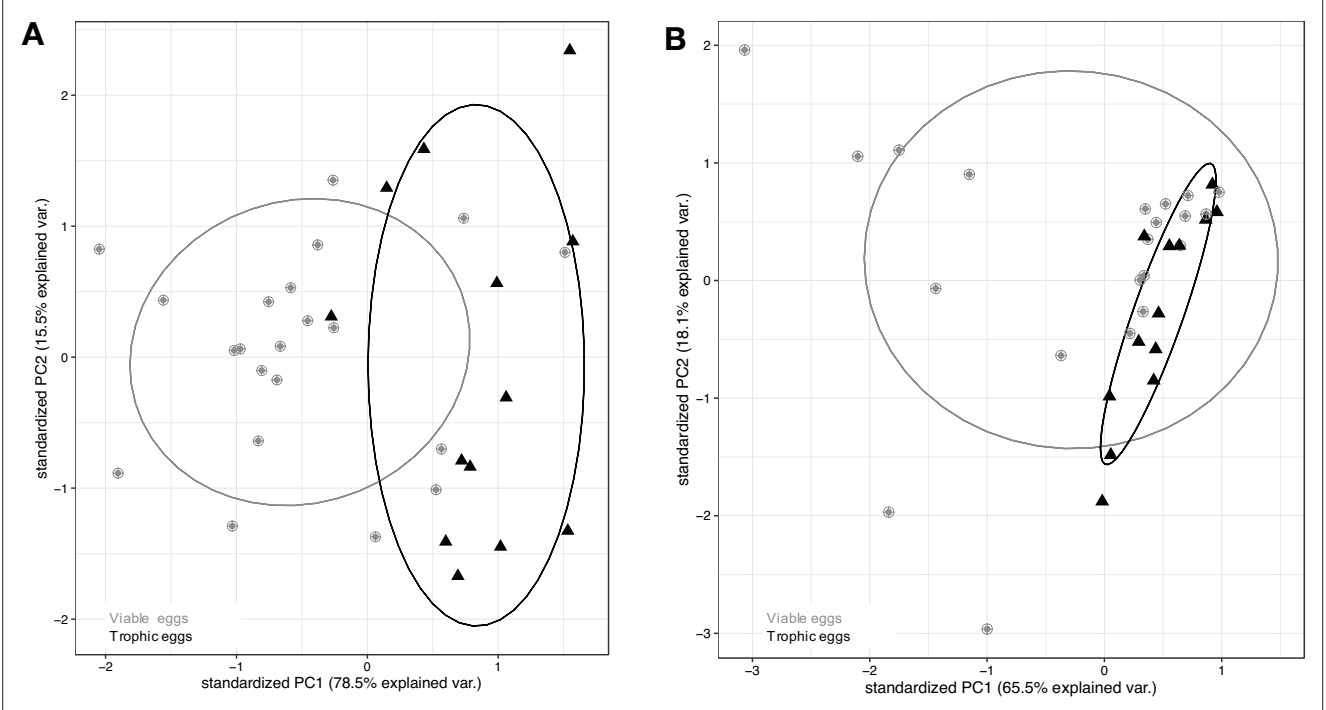

**Figure 5.** MiRNA and tRNA content of viable and trophic eggs. First two principal components (PC1 and PC2) explaining size distribution variation for (**A**) miRNA and (**B**) tRNA across egg samples, with viable eggs in gray dots and trophic eggs in black triangles. Ellipses enclose each of the egg type groups.

sealed with a cotton plug) at 28°C and 60% humidity, with a 12/12 hr light:dark cycle. They were fed ad libitum once a week with grass seeds, flies, and 20% honey water. Eggs were collected in October 2020 for the experiment investigating the effect of trophic eggs on larval caste fate, in November 2021 for estimating the percentage of trophic eggs and from February to December 2021 for the egg content analyses.

All statistical analyses were performed with RStudio (**RStudio Team, 2015**).

## Trophic and viable egg production

To verify that workers do not lay trophic eggs, as previously shown for other *Pogonomyrmex* species (**Supplementary file 1, table 1**), we created 12 queenless colonies (by removing the queen) and waited approximately 3 weeks until workers started laying eggs. From each of these colonies, we isolated two groups of five workers for 12 hr every 2 days for 2 weeks in November 2020 to obtain eggs. Collected eggs were then placed for 10 days in a Petri dish containing a water reservoir to study their development and distinguish whether they were trophic or viable.

To determine whether queens laid variable percentages of trophic eggs over time, we isolated each of 43 *P. rugosus* queens for 8 hr every day for 2 weeks, before and after hibernation, and counted the number of trophic and viable eggs laid (see results for how to discriminate the two types of eggs). We hibernated the queens because we previously showed that hibernation is important to trigger the production of gynes in *P. rugosus* colonies in the laboratory (**Schwander et al., 2008**; **Libbrecht et al., 2013**). Hibernation conditions were as described in **Libbrecht et al., 2013**. The percentage of trophic eggs was compared using a linear mixed-effects model with before vs after hibernation as the explanatory variable and colony as a random factor.

To assess whether viable and trophic eggs were laid in a random order, or whether eggs of a given type were laid in clusters, we isolated 11 queens for 10 hr, eight times over 3 weeks, and collected every hour the eggs laid. To determine whether viable and trophic eggs were laid in a random order, we performed a Wald–Wolfowitz runs test for each queen's egg-laying sequence (package *snpar* v.1.0; this non-parametric test calculates the likelihood that a binomial data sequence is random).

### Trophic egg influence on the larval caste fate

To determine whether trophic eggs influence the process of caste determination, we compared the development of freshly hatched (first instar) larvae placed in small recipient colonies with and without trophic eggs. From each of 22 donor colonies, we obtained approximately 30 freshly hatched larvae by isolating the queens for 16 hr (from 2 pm to 6 am) every day for 3 weeks (in October 2020), with a 24-hr break every 3 days. Eggs were collected every 8 hr and placed during 10 days in a Petri dish with a water reservoir ensuring a high humidity until they hatched. After hatching, for each colony, half of the larvae (i.e., 15 larvae) were then transferred into a recipient colony containing 20 workers, while the other half were placed in identical recipient colonies, which received in addition 45 0- to 4-hr-old trophic eggs (i.e., there were 3 trophic eggs per larva). There was no cross-fostering between colonies, so that larvae were always placed in recipient colonies containing workers from the same donor colony. The recipient colonies were maintained at 28°C and 60% humidity, with a 12/12 hr light:dark cycle and fed ad libitum twice a week with grass seeds, flies and 20% honey water. The caste of each newly produced individual was recorded at the pupal stage. To compare the proportion of queen pupae produced between recipient colonies with and without trophic eggs, we used the package *lme4* (*Bates et al., 2015*) to perform a binomial generalized linear mixed-effects analysis (GLMM/link='logit') fit by maximum likelihood, with caste as response variable (binary categorical factor) and presence/absence of trophic eggs as an explanatory variable. Donor colony was included as a random effect. To test whether the presence of trophic eggs affects survival, we performed a linear mixed-effects analysis with mortality as a response variable, presence/absence of trophic eggs as explanatory variable, and colonies as random effects. As we found a significantly higher survival of larvae in recipient colonies with trophic eggs than recipient colonies without trophic eggs (see results), we tested whether the percentage

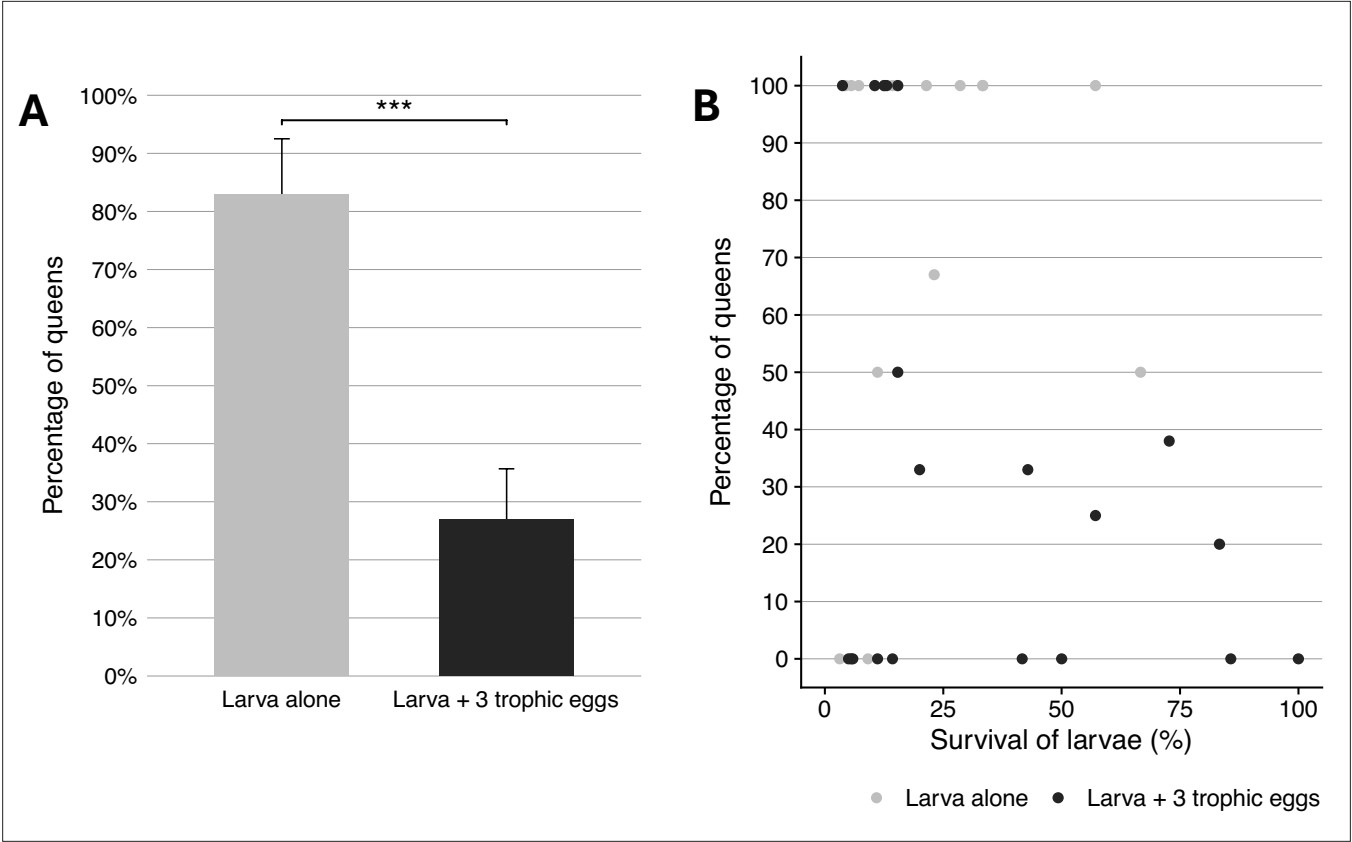

**Figure 6.** Caste fate and survival of larvae in the presence and absence of trophic eggs. (**A**) Percentage (± standard error) of queens among the larvae that developed to the pupal stage in colonies without (gray) or with (black) trophic eggs. (**B**) Relationship between the percentage of larvae who developed into queens and the survival of larvae (percentage) between the larval to pupal stages. *** indicated a p < 0.001.

of larvae developing into queens was correlated with survival by performing a linear mixed-effects analysis with the percentage of queen pupae as response variable, the survival as an explanatory variable and colonies as a random factor.

## Volume and content of trophic and viable eggs

The volumes of trophic ($n$ = 11) and viable eggs ($n$ = 14) were estimated by using the volume of an ellipse $\left(\frac{4}{3} \times \pi \times egg\ length \times \left(\frac{egg\ width}{2}\right)^2\right)$ with egg length and width estimated on images under ×10 magnification using ZEN Microscopy Software (v. 1.1.2.0).

To determine the nutritional content of viable and trophic eggs, we quantified the proteins, triglycerides, glycogen, and glucose in both types of eggs. We also quantified long and small RNAs (including miRNAs) as these compounds have been shown to be involved in caste determination in other eusocial species. To obtain the two types of eggs, we isolated 12 queens for 10 hr (7 am to 5 pm; from March to October 2021) in a dark Petri dish with three workers and a water supply. Eggs were collected every hour (so all eggs were a maximum of 1 hr old), and trophic and viable eggs were flash-frozen separately in liquid nitrogen. Twenty eggs were pooled for triglycerides–sugar–protein analyses and six eggs for RNA analyses. They were kept at –80°C until the extractions were performed. After the 10 hr of isolation, queens and workers were returned to their colony until the next isolation session. For each of the 12 colonies, we obtained two replicates of viable and trophic egg pools (i.e., 24 replicates in total).

Triglycerides, glycogen, and glucose were quantified as described in *Tennessen et al., 2014*, and protein levels were measured using a Bradford assay (*Bradford, 1976*). The 20 1-hr-old eggs per sample were homogenized with beads in 200 µl of PBS buffer in a Precellys Evolution tissue homogenizer coupled with a Cryolys Evolution (Bertin Technologies SAS).

For the Bradford assay, 10 µl of the homogenate was put in a clear-bottom 96-well plate with 300 µl of Coomassie Plus Reagent (Thermo Scientific: 23200) and incubated for 10 min at room temperature. Protein standard (Sigma: P5369) was used as standard (ranging from 0 to 0.5 mg/ml) and protein absorbance was read at 595 nm on a Hidex Sense Microplate Reader.

For the triglycerides assay, 90 µl of homogenate were heat treated at 70°C for 10 min, then 40 µl were mixed with 40 µl of Triglyceride Reagent (Sigma: T2449) for digestion and 40 µl were mixed with PBS buffer for free glycerol measurement. After 30-min incubation at 37°C, 30 µl of each sample and standards were transferred to a clear-bottom 96-well plate. 100 µl of Free Glycerol Reagent (Sigma: F6428) was added to each sample, mixed well by pipetting, and incubated 5 min at 37°C. Glycerol standard solution (Sigma: G7793) was used as standard (ranging from 0 to 1.0 mg/ml TAG) and absorbance was read at 540 nm on a Hidex Sense Microplate Reader. The triglycerides concentration in each sample was determined by subtracting the absorbance of free glycerol in the corresponding sample.

Glucose and glycogen were quantified as in *Tennessen et al., 2014*. A 90-µl aliquot was heat treated at 70°C for 10 min and then diluted 1:2 with PBS. The standard curves for glucose (Sigma, G6918) and glycogen (Sigma: G0885) were made by diluting stocks to 160 µg/ml, making 1:1 serial dilution for 160, 80, 40, 20, and 10 µg/ml. 40 µl of each sample was pipetted in duplicates of a clear microplate, and 30 µl of each glucose or glycogen standard was pipetted in duplicates. Amyloglucosidase enzyme (Sigma, A1602) was diluted 3 µl into 2000 µl of PBS, and 40 µl diluted enzyme was pipetted to the glycogen standards and to one well of the sample (for total glucose determination), 40 µl PBS was pipetted to the glucose standards and to the other sample well (for free glucose determination). The plate was incubated at 37°C for 60 min. 30 µl of each standard and samples (in duplicates) were transferred to a UV 96-well plate and 100 µl Glucose Assay Reagent (G3293) was pipetted to each well. The plate was incubated at room temperature for 15 min and the absorbance was read at 340 nm on a Hidex Sense Microplate Reader. The glycogen concentration was quantified by subtracting the free glucose absorbance from the total glycogen + glucose absorbance.

Concentrations of each compound (protein, triglycerides, glycogen, and glucose) were compared between viable and trophic eggs using a linear mixed-effects analysis (LMER; package *lme4*), with the concentration as response variable and egg type as explanatory variable. Colony and extraction batch were added as random effects in the model.

## Total and small RNA, and DNA

RNA (>200 nt) and small RNA were isolated using the miRNeasy Mini Kit (QIAGEN, cat. no. 217004) and RNeasy MinElute Cleanup Kit (QIAGEN, cat. no. 74204), respectively, following manufacturer instructions. RNA (>200 nt) and small RNA concentrations were measured with a QuantiFluor RNA System (Promega). RNA (>200 nt) integrity was examined with an Agilent Fragment Analyzer (at the Lausanne Genomic Technologies Facility) using a High Sensitivity Assay and small RNA were examined using the small RNA kit (at the Gene Expression Core Facility at EPFL).

The miRNA and RNA (>200 nt) concentrations were compared between viable and trophic eggs with paired $t$-tests (for each type of eggs, we used the average of the two replicates per colony). We also compared the fragment size distributions from 18 to 24 nucleotides for miRNAs (*Sohel, 2016*) with a PCA and a Mantel test.

DNA was extracted from pools of six eggs using TRIzol (Life Technologies). DNA concentration was measured with a Nanodrop 3300 (Thermo Fisher), and DNA integrity was examined with an Agilent Fragment Analyzer (at the Lausanne Genomic Technologies Facility) using a High Sensitivity Assay. DNA concentrations were compared between viable and trophic eggs using paired $t$-tests (sample size is 5 for both types of eggs, each sample being a pool of 6 eggs).

## Acknowledgements

We thank Dr C Berney for technical assistance to develop wet lab protocols. We are grateful to S McGregor and M Chapuisat for their helpful comments on the manuscript. This work was supported by an ERC grant and the Swiss NSF (LK) and funding from the University of Lausanne (LK and TS).

## Additional information

### Competing interests

Laurent Keller: Reviewing editor, eLife. The other authors declare that no competing interests exist.

### Funding

| Funder | Grant reference number | Author |
|---|---|---|
| Swiss National Science Foundation | 310030_200437 | Laurent Keller |
| University of Lausanne | | Laurent Keller Tanja Schwander |
| European Research Council | 10.3030/741491 | Laurent Keller |

The funders had no role in study design, data collection, and interpretation, or the decision to submit the work for publication.

### Author contributions

Eléonore Genzoni, Data curation, Formal analysis, Writing – original draft; Tanja Schwander, Conceptualization, Resources, Supervision, Methodology, Project administration, Writing – review and editing; Laurent Keller, Conceptualization, Formal analysis, Supervision, Funding acquisition, Investigation, Writing – original draft, Project administration, Writing – review and editing

### Author ORCIDs

Eléonore Genzoni (iD) https://orcid.org/0000-0003-3152-992X
Laurent Keller (iD) https://orcid.org/0000-0002-5046-9953

Reviewer #1 (Public Review): https://doi.org/10.7554/eLife.86899.4.sa1
Reviewer #2 (Public review): https://doi.org/10.7554/eLife.86899.4.sa2
Author response https://doi.org/10.7554/eLife.86899.4.sa3

# Additional files

## Supplementary files
Supplementary file 1. Supplementary tables.

MDAR checklist

## Data availability
All data generated or analyzed during this study are included in the manuscript and supporting files. For *Figure 1*, all the details are in *Supplementary file 1, table 1*.

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
