## [Editor Report · eLife Assessment]

This **important** manuscript by Genzoni et al. reports the striking discovery of a regulatory role for trophic eggs in ant caste determination. Prior to this study, trophic eggs were widely assumed to play only a nutritional role in the colony, but this **compelling** study shows that trophic eggs can suppress queen development, and therefore regulate caste determination in specific social contexts.

---

## [Referee Report · Reviewer #1 (Public Review)]

This manuscript describes a series of experiments documenting trophic egg production in a species of harvester ant, Pogonomyrmex rugosus. In brief, queens are the primary trophic egg producers, there is seasonality and periodicity to trophic egg production, trophic eggs differ in many basic dimensions and contents relative to reproductive eggs, and diets supplemented with trophic eggs had an effect on the queen/worker ratio produced (increasing worker production).

The manuscript is very well prepared and the methods are sufficient. The outcomes are interesting and help fill gaps in knowledge, both on ants as well as insects, more generally.

---

## [Referee Report · Reviewer #2 (Public review)]

The revised manuscript by Genzoni et al. reports the striking discovery of a regulatory role for trophic eggs. Prior to this study, trophic eggs were widely assumed to play a nutritional role in the colony, but this study shows that trophic eggs can suppress queen development, and therefore, can play a role in regulating caste determination in specific social contexts. In this revised version of the manuscript, the authors have addressed many of the concerns raised in the first version regarding the lack of sufficient information and context in the Introduction and Discussion.

---

## [Author Response]

The following is the authors’ response to the previous reviews

**Public Reviews:**

**Reviewer #1 (Public Review):**
This manuscript describes a series of experiments documenting trophic egg production in a species of harvester ant, Pogonomyrmex rugosus. In brief, queens are the primary trophic egg producers, there is seasonality and periodicity to trophic egg production, trophic eggs differ in many basic dimensions and contents relative to reproductive eggs, and diets supplemented with trophic eggs had an effect on the queen/worker ratio produced (increasing worker production).The manuscript is very well prepared and the methods are sufficient. The outcomes are interesting and help fill gaps in knowledge, both on ants as well as insects, more generally. More context could enrich the study and flow could be improved.

We thank the reviewer for these comments. We agree that the paper would benefit from more context. We have therefore greatly extended the introduction.

**Reviewer #2 (Public Review):**
The manuscript by Genzoni et al. provides evidence that trophic eggs laid by the queen in the ant Pogonomyrmex rugosis have an inhibitory effect on queen development. The authors also compare a number of features of trophic eggs, including protein, DNA, RNA, and miRNA content, to reproductive eggs. To support their argument that trophic eggs have an inhibitory effect on queen development, the authors show that trophic eggs have a lower content of protein, triglycerides, glycogen, and glucose than reproductive eggs, and that their miRNA distributions are different relative to reproductive eggs. Although the finding of an inhibitory influence of trophic eggs on queen development is indeed arresting, the egg cross-fostering experiment that supports this finding can be effectively boiled down to a single figure (Figure 6). The rest of the data are supplementary and correlative in nature (and can be combined), especially the miRNA differences shown between trophic and reproductive eggs. This means that the authors have not yet identified the mechanism through which the inhibitory effect on queen development is occurring. To this reviewer, this finding is more appropriate as a short report and not a research article. A full research article would be warranted if the authors had identified the mechanism underlying the inhibitory effect on queen development. Furthermore, the article is written poorly and lacks much background information necessary for the general reader to properly evaluate the robustness of the conclusions and to appreciate the significance of the findings.

We thank the reviewer for these comments. We agree that the paper would benefit by having more background information and more discussion. We have followed this advice in the revision.

**Reviewer #3 (Public Review):**
In "Trophic eggs affect caste determination in the ant Pogonomyrmex rugosus" Genzoni et al. probe a fundamental question in sociobiology, what are the molecular and developmental processes governing caste determination? In many social insect lineages, caste determination is a major ontogenetic milestone that establishes the discrete queen and worker life histories that make up the fundamental units of their colonies. Over the last century, mechanisms of caste determination, particularly regulators of caste during development, have remained relatively elusive. Here, Genzoni et al. discovered an unexpected role for trophic eggs in suppressing queen development - where bi-potential larvae fed trophic eggs become significantly more likely to develop into workers instead of gynes (new queens). These results are unexpected, and potentially paradigm-shifting, given that previously trophic eggs have been hypothesized to evolve to act as an additional intracolony resource for colonies in potentially competitive environments or during specific times in colony ontogeny (colony foundation), where additional food sources independent of foraging would be beneficial. While the evidence and methods used are compelling (e.g., the sequence of reproductive vs. trophic egg deposition by single queens, which highlights that the production of trophic eggs is tightly regulated), the connective tissue linking many experiments is missing and the downstream mechanism is speculative (e.g., whether miRNA, proteins, triglycerides, glycogen levels in trophic eggs is what suppresses queen development). Overall, this research elevates the importance of trophic eggs in regulating queen and worker development but how this is achieved remains unknown.

We thank the reviewer for these comments and agree that future work should focus on identifying the substances in trophic eggs that are responsible for caste determination.

**Recommendations For The Authors:**

**Reviewer #1 (Recommendations For The Authors):**
Introduction:The context for this study is insufficiently developed in the introduction - it would be nice to have a more detailed survey of what is known about trophic eggs in insects, especially social insects. The end of the introduction nicely sets up the hypothesis through the prior work described by Helms Cahan et al. (2011) where they found JH supplementation increased trophic egg production and also increased worker size. I think that the introduction could give more context about egg production in Pogonomyrmex and other ants, including what is known about worker reproduction. For example, Suni et al. 2007 and Smith et al. 2007 both describe the absence of male production by workers in two different harvester ants. Workers tend to have underdeveloped ovaries when in the presence of the queen. Other species of ants are known to have worker reproduction seemingly for the purpose of nutrition (see Heinze and Hölldober 1995 and subsequent studies on Crematogaster smithi). Because some ants, including Pogonomyrmex, lack trophallaxis, it has been hypothesized that they distribute nutrients throughout the nest via trophic eggs as is seen in at least one other ant (Gobin and Ito 2000). Interestingly, Smith and Suarez (2009) speculated that the difference in nutrition of developing sexual versus worker larvae (as seen in their pupal stable isotope values) was due to trophic egg provisioning - they predicted the opposite as was found in this study, but their prediction was in line with that of Helms Cahan et al. (2011). This is all to say that there is a lot of context that could go into developing the ideas tested in this paper that is completely overlooked. The inclusion of more of what is known already would greatly enrich the introduction.

We agree that it would be useful to provide a larger context to the study. We now provide more information on the life-history of ants and explained under what situations queens and workers may produce trophic eggs. We also mentioned that some ants such as *Crematogaster smithi* have a special caste of “large workers” which are morphologically intermediate between winged queens and small workers and appear to be specialized in the production of unfertilized eggs. We now also mention the study of Goby and Ito (200) where the authors show that trophic eggs may play an important role in food distribution withing the colony, in particular in species where trophallaxis is rare or absent.

Methods:L49: What lineage is represented in the colonies used? The collection location is near where both dependent-lineage (genetic caste determining) P. rugosus and "H" lineage exist. This is important to know. Further, depending on what these are, the authors should note whether this has relevance to the study. Not mentioning genetic caste determination in a paper that examines caste determination is problematic.

This is a good point. We have now provided information at the very beginning of the material and method section that the queens had been collected in populations known not to have dependentlineage (genetic caste determining) mechanisms of caste determination.

L63 and throughout: It would be more efficient to have a paragraph that cites R (must be done) and RStudio once as the tool for all analyses. It also seems that most model construction and testing was done using lme4 - so just lay this out once instead of over and over.

We agree and have updated the manuscript accordingly.

L95: 'lenght' needs to be 'length' in the formula.

Thanks, corrected.

L151: A PCA was used but not described in the methods. This should be covered here. And while a Mantel test is used, I might consider a permANOVA as this more intuitively (for me, at least) goes along with the PCA.

We added the PCA description in the Material and Method section.

Results:I love Fig. 3! Super cool.

Thanks for this positive comment.

Discussion:It would be good to have more on egg cannibalism. This is reasonably well-studied and could be good extra context.

We have added a paragraph in the discussion to mention that egg cannibalism is ubiquitous in ants.

Supp Table 1: P. badius is missing and citations are incorrectly attributed to P. barbatus.

*P. badius* was present in the Table but not with the other *Pogonomyrmex* species. For some genera the species were also not listed in alphabetic order. This has been corrected.

**Reviewer #2 (Recommendations For The Authors):**
Comments on introduction:The introduction is missing information about caste determination in ants generally and Pogonomyrmex rugosis specifically. This is important because some colonies of Pogonomyrmex rugosis have been shown to undergo genetic caste determination, in which case the main result would be rendered insignificant. What is the evidence that caste determination in the lineages/colonies used is largely environmentally influenced and in what contexts/environmental factors? All of this should be made clear.

This is a good point. We have expanded the introduction to discuss previous work on caste determination in *Pogonomyrmex* species with environmental caste determination and now also provide evidence at the beginning of the Material and Method section that the two populations studied do not have a system of genetic caste determination.

Line 32 and throughout the paper: What is meant exactly by 'reproductive eggs'? Are these eggs that develop specifically into reproductives (i.e., queens/males) or all eggs that are non-trophic? If the latter, then it is best to refer to these eggs as 'viable' in order to prevent confusion.

We agree and have updated the manuscript accordingly.

Figure 1/Supp Table 1: It is surprising how few species are known to lay trophic eggs. Do the authors think this is an informative representation of the distribution of trophic egg production across subfamilies, or due to lack of study? Furthermore, the branches show ant subfamilies, not families. What does the question mark indicate? Also, the information in the table next to the phylogeny is not easy to understand. Having in the branches that information, in categories, shown in color for example, could be better and more informative. Finally, having the 'none' column with only one entry is confusing - discuss that only one species has been shown to definitely not lay trophic eggs in the text, but it does not add much to the figure.

Trophic eggs are probably very common in ants, but this has not been very well studied. We added a sentence in the manuscript to make this clear.

Thanks for noticing the error family/subfamily error. This has been corrected in Figure 1 and Supplementary Table 1.

The question mark indicates uncertainty about whether queens also contribute to the production of trophic eggs in one species (*Lasius niger*). We have now added information on that in the Figure legend.

We agree with the reviewer that it would be easier to have the information on whether queens and workers produce trophic on the branches of the Tree. However, having the information on the branches would suggest that the “trait” evolved on this part of the tree. As we do not know when worker or queen production of trophic eggs exactly evolved, we prefer to keep the figure as it is.

Finally, we have also removed the none in the figure as suggested by the reviewer and discussed in the manuscript the fact that the absence of trophic eggs has been reported in only one ant species (*Amblyopone silvestrii*: Masuko 2003_)._

Comments on materials and methods:Why did they settle on three trophic eggs per larva for their experimental setup?

We used three trophic eggs because under natural conditions 50-65% of the eggs are trophic. The ratio of trophic eggs to viable eggs (larvae) was thus similar natural condition.

Line 50: In what kind of setup were the ants kept? Plaster nests? Plastic boxes? Tubes? Was the setup dry or moist? I think this information is important to know in the context of trophic eggs.

We now explain that colonies were maintained in plastic boxes with water tubes.

Line 60: Were all the 43 queens isolated only once, or multiple times?

Each of the 43 queens were isolated for 8 hours every day for 2 weeks, once before and once after hibernation (so they were isolated multiple times). We have changed the text to make clear that this was done for each of the 43 queens.

Could isolating the queen away from workers/brood have had an effect on the type of eggs laid?

This cannot be completely ruled out. However, it is possible to reliably determine the proportion of viable and trophic eggs only by isolating queens. And importantly the main aim of these experiments was not to precisely determine the proportion viable and trophic eggs, but to show that this proportion changes before and after hibernation and that queens do not lay viable and trophic eggs in a random sequence.

Since it was established that only queens lay trophic eggs why was the isolation necessary?

Yes this was necessary because eggs are fragile and very difficult to collect in colonies with workers (as soon as eggs are laid they are piled up and as soon as we disturb the nest, a worker takes them all and runs away with them). Moreover, it is possible that workers preferentially eat one type of eggs thus requiring to remove eggs as soon as queens would have laid them. This would have been a huge disturbance for the colonies.

Line 61: Is this hibernation natural or lab induced? What is the purpose of it? How long was the hibernation and at what temperature? Where are the references for the requirement of a diapause and its length?

The hibernation was lab induced. We hibernated the queens because we previously showed that hibernation is important to trigger the production of gynes in *P. rugosus* colonies in the laboratory (Schwander et al 2008; Libbrecht et al 2013). Hibernation conditions were as described in Libbrecht et al (2013).

Line 73: If the queen is disturbed several times for three weeks, which effect does it have on its egg-laying rate and on the eggs laid? Were the eggs equally distributed in time in the recipient colonies with and without trophic eggs to avoid possible effects?

It is difficult to respond what was the effect of disturbance on the number and type of eggs laid. But again our aim was not to precisely determine these values but determine whether there was an effect of hibernation on the proportion of trophic eggs. The recipient colonies with and without trophic eggs were formed in exactly the same way. No viable eggs were introduced in these colonies, but all first instar larvae have been introduced in the same way, at the same time, and with random assignment. We have clarified this in the Material and Method section.

Line 77: Before placing the freshly hatched larvae in recipient colonies, how long were the recipient colonies kept without eggs and how long were they fed before giving the eggs? Were they kept long enough without the queen to avoid possible effects of trophic eggs, or too long so that their behavior changed?

The recipient colonies were created 7 to 10 days before receiving the first larvae and were fed ad libitum with grass seeds, flies and honey water from the beginning. Trophic eggs that would have been left over from the source colony should have been eaten within the first few days after creating the recipient colonies. However, even if some trophic eggs would have remained, this would not influence our conclusion that trophic eggs influence caste fate, given the fully randomized nature of our treatments and the considerable number of independent replicates. The same applies to potential changes in worker behavior following their isolation from the queen.

Line 77: Is it known at what stage caste determination occurs in this species? Here first instar larvae were given trophic eggs or not. Does caste-determination occur at the first instar stage? If not, what effect could providing trophic eggs at other stages have on caste-determination?

A previous study showed that there is a maternal effect on caste determination in the focal species (Schwander et al 2008). The mechanism underlying this maternal effect was hypothesized to be differential maternal provisioning of viable eggs. However, as we detail in the discussion, the new data presented in our study suggests that the mechanism is in fact a different abundance of trophic eggs laid by queens. There is currently no information when exactly caste determination occurs during development

Comments on results:Line 65: How does investigating the order of eggs laid help to "inform on the mechanisms of oogenesis"?

We agree that the aim was not to study the mechanism of oogenesis. We have changed this sentence accordingly: “To assess whether viable and trophic eggs were laid in a random order, or whether eggs of a given type were laid in clusters, we isolated 11 queens for 10 hours, eight times over three weeks, and collected every hour the eggs laid”

Figure 2: There is no description/discussion of data shown in panels B, C, E, and F in the main text.

We have added information in the main text that while viable eggs showed embryonic development at 25 and 65 hours (Fig 12 B, C) there was no such development for trophic eggs (Fig. 2 E,F).

Line 172: Please explain hibernation details and its significance on colony development/life cycle.

We have added this information in the Material and Method section.

Figure 6: How is B plotted? How could 0% of gynes have 100% survival?

The survival is given for the larvae without considering caste. We have changed the de X axis of panel B and reworded the Figure legend to clarify this.

Is reduced DNA content just an outcome of reduced cell number within trophic eggs, i.e., was this a difference in cell type or cell number? Or is it some other adaptive reason?

It is likely to be due to a reduction in cell number (trophic eggs have maternal DNA in the chorion, while viable eggs have in addition the cells from the developing zygote) but we do not have data to make this point.

Is there a logical sequence to the sequence of egg production? The authors showed that the sequence is non-random, but can they identify in what way? What would the biological significance be?

We could not identify a logical sequence. Plausibly, the production of the two types of eggs implies some changes in the metabolic processes during egg production resulting in queens producing batches of either viable or trophic eggs. This would be an interesting question to study, but this is beyond the scope of this paper.

Figure 6b is difficult to follow, and more generally, legends for all figures can be made clearer and more easy to follow.

We agree. We have now improved the legends of Fig 6B and the other figures.

Lines 172-174: "The percentage of eggs that were trophic was higher before hibernation...than after. This higher percentage was due to a reduced number of reproductive eggs, the number of trophic eggs laid remained stable" - are these data shown? It would be nice to see how the total egglaying rate changes after hibernation. Also, is the proportion of trophic eggs laid similar between individual queens?

No the data were not shown and we do not have excellent data to make this point. We have therefore removed the sentence “This higher percentage was due to a reduced number of reproductive eggs, the number of trophic eggs laid remained stable” from the manuscript.

Figure 6B: Do several colonies produce 100% gynes despite receiving trophic eggs? It would be interesting if the authors discussed why this might occur (e.g., the larvae are already fully determined to be queens and not responsive to whatever signal is in the trophic eggs).

The reviewer is correct that 4 colonies produced 100% gynes despite receiving trophic eggs. However, the number of individuals produced in these four colonies was small (2,1,2,1, see supplementary Table 2). So, it is likely that it is just by chance that these colonies produced only gynes.

Figure 5: Why a separation by "size distribution variation of miRNA"? What is the relevance of looking at size distributions as opposed to levels?

We did that because there many different miRNA species, reflected by the fact that there is not just one size peak but multiple one. This is why we looked at size distribution

Figure 2: The image of the viable embryo is not clear. If possible, redo the viable to show better quality images.

Unfortunately, we do not anymore have colonies in the laboratory so this is not possible.

Comments on discussion:Lines 236-247: Can an explanation be provided as to why the effect of trophic eggs in P. rugosus is the opposite of those observed by studies referenced in this section? Could P. rugosus have any life history traits that might explain this observation?

In the two mentioned studies there were other factors that co-varied with variation in the quantity of trophic eggs. We mentioned that and suggested that it would be useful to conduct experimental manipulation of the quantity of trophic eggs in the Argentine ant and *P. barbatus* (the two species where an effect of trophic eggs had been suggested).

The discussion should include implications and future research of the discovery.

We made some suggestions of experiments that should be performed in the future

The conclusion paragraph is too short and does not represent what was discussed.

We added two sentences at the end of the paragraph to make suggestions of future studies that could be performed.

Lines 231 to 247: Drastically reduce and move this whole part to the introduction to substantiate the assumption that trophic eggs play a nutritional role.

We moved most of this paragraph to the introduction, as suggested by the reviewer.

**Reviewer #3 (Recommendations For The Authors):**
I would like to commend the authors on their study. The main findings of the paper are individually solid and provide novel insight into caste determination and the nature of trophic eggs. However, the inferences made from much of the data and connections between independent lines of evidence often extend too far and are unsubstantiated.

We thank the reviewer for the positive comment. We made many changes in the manuscript to improve the discussion of our results.